# Activated heme synthesis regulates glycolysis and oxidative metabolism in breast and ovarian cancer cells

**Pritpal Kaur, Shreya Nagar, Madhura Bhagwat, Mohammad Uddin, Yan Zhu, Ivana Vancurova, Ales Vancura** *

Department of Biological Sciences, St. John's University, New York, United States of America

* vancuraa@stjohns.edu

**Data Availability Statement:** All relevant data are within the paper.

**Funding:** AV: National Institutes of Health grant GM135839,GM120710; URL: www.nih.gov The

## Abstract

Heme is an essential cofactor for enzymes of the electron transport chain (ETC) and ATP synthesis in mitochondrial oxidative phosphorylation (OXPHOS). Heme also binds to and destabilizes Bach1, a transcription regulator that controls expression of several groups of genes important for glycolysis, ETC, and metastasis of cancer cells. Heme synthesis can thus affect pathways through which cells generate energy and precursors for anabolism. In addition, increased heme synthesis may trigger oxidative stress. Since many cancers are characterized by a high glycolytic rate regardless of oxygen availability, targeting glycolysis, ETC, and OXPHOS have emerged as a potential therapeutic strategy. Here, we report that enhancing heme synthesis through exogenous supplementation of heme precursor 5-amino-levulinic acid (ALA) suppresses oxidative metabolism as well as glycolysis and significantly reduces proliferation of both ovarian and breast cancer cells. ALA supplementation also destabilizes Bach1 and inhibits migration of both cell types. Our data indicate that the underlying mechanisms differ in ovarian and breast cancer cells, but involve destabilization of Bach1, AMPK activation, and induction of oxidative stress. In addition, there appears to be an inverse correlation between the activity of oxidative metabolism and ALA sensitivity. Promoting heme synthesis by ALA supplementation may thus represent a promising new anti-cancer strategy, particularly in cancers that are sensitive to altered redox signaling, or in combination with strategies that target the antioxidant systems or metabolic weaknesses of cancer cells.

## Introduction

Cancer cells undergo a complex rearrangement of metabolic pathways involved in biosynthetic processes to support tumor initiation and progression [1]. Altered metabolism has long been considered as a consequence of cell transformation rather than an active participant in tumorigenesis. However, more recent cancer metabolism studies provide many examples of metabolic activities that actively contribute to cell transformation and promote tumorigenesis [1]. One of the hallmarks of cancer is a high rate of glucose uptake and lactate production regardless of oxygen availability, referred to as the "Warburg effect" [2]. The high glycolytic rate

funders had no role in study design, data collection and analysis, decision to publish, or preparation of the manuscript.

**Competing interests:** The authors have declared that no competing interests exist.

supports the biosynthetic needs of proliferating cancer cells by providing intermediates of the glycolytic pathway for anabolic reactions. In addition, many tumor cells overexpress genes encoding metabolic enzymes that facilitate synthesis of ribose 5-phosphate, NADPH, fatty acids, and glycerophospholipids [3–6]. Despite the diversion of carbon flux away from the tricarboxylic acid (TCA) cycle, electron transport chain (ETC), and oxidative phosphorylation (OXPHOS) pathways, most tumors contain functional mitochondria. However, due to the decreased metabolic flux through the TCA cycle in many tumors, intermediates of the TCA cycle can become limiting for synthesis of certain amino acids and cancer cell proliferation [7, 8]. Thus, targeting glycolysis, ETC, OXPHOS, and TCA cycle have emerged as a potential therapeutic strategy in some cancers [9–16].

Heme has a potential to regulate glycolysis, ETC, OXPHOS, and TCA cycle [17]. Heme is an essential cofactor of several enzymes of the ETC and is required for ATP production by the mitochondrial ETC and OXPHOS. Increased heme synthesis is associated with disruption of the Warburg effect and increased ETC and OXPHOS [18–20]. In yeast cells, increased synthesis of heme induces oxidative metabolism [21]. On the other hand, free heme that is not incorporated into hemoproteins catalyzes production of free radicals through Fenton chemistry [22] and elevated synthesis of heme or accumulation of intermediates in heme biosynthetic pathway may trigger increased synthesis of ROS [23]. In addition, cellular heme level regulates the stability of Bach1 (BTB domain and CNC homolog 1). Bach1 is a transcriptional regulator that activates expression of glycolytic genes and represses transcription of the TCA cycle and ETC genes in lung and breast cancer cells [24–26]. In addition, Bach1 activates transcription of genes important for cell migration and promotes metastasis [27–29]. Heme binds to Bach1 and stimulates its degradation [30]. Under normal redox conditions, Bach1 binds to antioxidant response elements (AREs) in the promoters of antioxidant genes, such as heme oxygenase -1 (HO-1), and represses their transcription. Oxidative stress elevates intracellular level of free heme by releasing heme from hemoproteins, leading to degradation of Bach1 and transcriptional induction of antioxidant genes, such as catalase. Stabilization of Bach1 was recently implicated in lung cancer metastasis [24, 25]. The level of Bach1 is also elevated in triple-negative breast cancer (TNBC), leading to lower activity of TCA cycle and decreased transcription of ETC genes. Suppressing or destabilizing Bach1 renders TNBC cells sensitive to ETC inhibitors, such as metformin [26].

In this study, we explored the role of cellular heme level in regulation of Warburg effect and expression of pro-metastatic genes in ovarian cancer and TNBC cells. Additional motivation for this study was to determine whether supplementation with heme precursor 5-aminolevulinic acid (ALA) downregulates Bach1. We show that ALA supplementation leads to downregulation of glycolysis, mitochondrial respiration, as well as cell migration. The underlying mechanisms differ in ovarian and TNBC cells, but likely involve destabilization of Bach1, activation of AMP-activated protein kinase (AMPK), and induction of anti-oxidant response.

## Experimetal procedures

### Reagents

5-Aminolevulinic acid hydrochloride (ALA, A3785) was purchased from Sigma. 5-Amino-1-β-D-ribofuranosyl-1H-imidazole-4-carboxamide (AICAR, 10010241) was purchased from Cayman.

### Cell culture

All cell lines were obtained from American Type Culture Collection (ATCC, Manassas, VA, USA). MCF7, H1299 and MDA-MB-231 cells were cultured in Dulbecco's modified Eagle's

medium (DMEM) supplemented with 10% heat inactivated fetal bovine serum (FBS; Invitrogen, Grand Island, NY, USA) and antibiotics (100 units/mL penicillin and 100 μg/ml streptomycin). MDA-MB-468 and SKOV3 cells were cultured in RPMI 1640 medium (Invitrogen) supplemented with 2 mM L-glutamine, 10 mM HEPES, 1 mM sodium pyruvate, 10% heat inactivated FBS, and antibiotics (100 units/ml penicillin and 100 μg/ml streptomycin). OVCAR3 cells were cultured in RPMI 1640 medium (Invitrogen) supplemented with 2 mM L-glutamine, 10 mM HEPES, 1 mM sodium pyruvate, 20% heat inactivated FBS, and antibiotics (100 units/ml penicillin and 100 μg/ml streptomycin). Cells were seeded ($5 \times 10^5$ cells/ml) for 24 h in 6-well plates, grown at 37˚C with 5% CO2, and incubated with ALA or AICAR for 24 h. Cell viability was measured using Trypan Blue exclusion. Cell proliferation was measured by CellTiter96 One Solution Cell Proliferation Assay (Promega, Madison, WI) as described [31].

## Transfection with siRNA

Human Bach1 (sc-37064), and non-silencing (sc-37007) small interfering RNAs (siRNAs) were obtained from Santa Cruz Biotechnology. Prior to transfection, cells were seeded ($2 \times 10^5$ cells/ml) into a 6-well plate and incubated in a humidified 5% $CO_2$ atmosphere at 37˚C in antibiotic free medium supplement with 10% FBS for 24 h to about 80% confluence. For each transfection, 80 pmol of either non-silencing siRNA control or Bach1 siRNA were used; cells were transfected 7 h in siRNA transfection medium with siRNA transfection reagent according to manufacturer's instructions (Santa Cruz Biotechnology). After transfection, fresh medium with antibiotics was added, and cells were grown for 24 h.

## Wound-healing scratch assay

Cells were seeded in 6-well plates at a density of $5 \times 10^5$ cells/ml in medium containing 5% FBS. Once the cells became confluent, a wound area was created by scraping the cell monolayer with a sterile 200 μl pipette tip. After washing twice with PBS, DMEM medium without FBS or RPMI 1640 medium without FBS was added to the wells with MDA MB 231 or SKOV3 cells, respectively. The scratch area was monitored under a phase-contrast microscope at 0, 24, 48 and 72 h after transfection. The wound width was measured in five random fields using ImageJ software as described [32] and the average wound width at 72 h was normalized to the average wound width at 0 h.

## Heme assay

The amount of heme was measured using heme assay kit (K672-100, BioVision) according to the manufacturer's instruction. Cells were seeded into 6-well plates at a density of $5 \times 10^5$ cells/ml in culture medium. After overnight incubation, cells were treated with ALA and AICAR and incubated for 24 h at 37˚C in humidified air with 5% $CO_2$. After 24 h treatment, cells were harvested, washed twice with PBS, and lysed in 50 μl of 1% NaOH and 5% Triton. Cell lysates were neutralized to pH 7.5 and used to measure the amount of heme.

## Lactate assay

Lactate concentration in the culture medium was measured using lactate assay kit (MAK065, Millipore-Sigma) according to the manufacturer's instruction. Cells were seeded into 6-well plates at a density of $5 \times 10^5$ cells/ml in culture medium. After overnight incubation, cells were treated with ALA or AICAR, and incubated for 24 h at 37˚C in humidified air with 5% $CO_2$. After 24 h treatment, cell medium was used to measure lactate concentration.

## Oxygen consumption rate (OCR) and extracellular acidification rate (ECAR)

OCR and ECAR were measured using Seahorse XFp analyzer as described previously [33]. Cells were plated in Seahorse XFp cell culture plates (103022, Agilent) at 20,000 cells/well and incubated overnight. After incubation, cells were treated with ALA and AICAR for 24 h. Cells were washed and incubated with assay medium at 37°C for 1 h in a non-$CO_2$ incubator. OCR and ECAR were measured using Seahorse XFp mito stress kit (103010, Agilent). OCR was measured under basal conditions followed by the sequential addition of oligomycin (1.5 μM), FCCP (0.5 μM), and rotenone & antimycin A (Rot/AA; 0.5 μM). Derived respiratory parameters are basal respiration (B), proton leak (PL), maximal respiration (MR), non-mitochondrial respiration (NMR), and ATP production (ATPP). For Bach1 suppressed cells, Bach1 siRNA transfection was done as described above. After transfection, cells were seeded in Seahorse XFp cell culture plates and assay were performed as described above.

## Real-time RT-PCR

Total RNA was isolated using RNeasy mini-kit (Qiagen, Valencia, CA). The iScript one-step RT-PCR kit with SYBR Green (Bio-Rad) was used as a supermix and 20 ng/μl of RNA was used as template on a Bio-Rad MyIQ Single Color Real-time PCR Detection System (Bio-Rad). The primers used for mRNA quantification are listed in Table 1. The results were normalized to 18S ribosomal RNA and the mRNA values were expressed as a percentage of control or untreated samples, which were arbitrarily set as 100%.

## Western blot

Whole cell extracts were prepared and western blotting was performed as described previously [34, 35]. The following primary antibodies were used: anti AMPK monoclonal antibody (2603; Cell Signaling) at a dilution of 1:1000; anti phospho-AMPK (Thr172) monoclonal antibody (pAMPK; 2535; Cell Signaling) at a dilution of 1:1000; anti-acetyl-CoA carboxylase

**Table 1. Primer sequences.**

| Gene | Forward sequence (5' to 3') | Reverse sequence (5' to 3') |
|------|------------------------------|------------------------------|
| 18S | CGGCTACCACATCCAAGGAA | GCTGGAATTACCGCGGCT |
| Bach1 | TGCGATGTCACCATCTTTGT | CCTGGCCTACGATTCTTGAG |
| HO-1 | AAGACTGCGTTCCTGCTCAAC | AAGCCCTACAGCAACTGTCG |
| HK2 | GAGCCACCACTCACCCTACT | CCAGGCATTCGGCAATGTG |
| GAPDH | GGAGCGAGATCCCTCCAAAAT | GGCTGTTGTCATACTTCTCATGG |
| NDUFB5 | AGCTGGAAGTGCGAAAATTGA | ATACCAGGGTCCATCTCCTCT |
| COX15 | CAGCGCCTAGAGCACAGTG | GCCAGACTCTGTCAACCTAGT |
| COX17 | TGCGTGTATCATCGAGAAAGGA | GCCTCAATTAGATGTCCACAGTG |
| ATP5D | CCCTCCTTGGTCAAGAGCAC | GTATCTCCGGTCGTTTCAGCA |
| SLC25A15 | CCTGAAGACTTACTCCCAGGT | GCGATGTTGGCGATTAGTGC |
| Catalase | TGGAGCTGGTAACCCAGTAGG | CCTTTGCCTTGGAGTATTTGGTA |
| MMP1 | GCGCACAAATCCCTTCTACC | ATCCGTGTAGCACATTCTGTCC |
| MMP9 | ATTTCTGCCAGGACCGCTTCTAC | ATCCGGCAAACTGGCTCCTTC |
| MMP13 | TGCAGAGCGCTACCTGAGATCATAC | GGAGCTTGCTGCATTCTCCTTCA |
| CXCR4 | TCTTCCTGCCCACCATCTACTC | TGCAGCCTGTACTTGTCCGTC |
| Vimentin | CAGGCAAAGCAGGAGTCCA | AAGTTCTCTTCCATTTCACGCA |

monoclonal antibody (ACCA; 3676; Cell Signaling) at a dilution of 1:1000; anti-phospho ace-tyl-CoA carboxylase (Ser79) polyclonal antibody (pACCA; 3661; Cell Signaling) at a dilution of 1:1000; anti-Bach1 monoclonal antibody (Bach1; sc-271211; Santa Cruz Biotechnology) at a dilution of 1:1000; anti-actin polyclonal antibody (A5060, Sigma) at a dilution of 1:2000.

## Statistical analysis

The results represent at least three independent experiments. Numerical results are presented as means ± SD. Data were analyzed by using an InStat software package (GrapPAD, San Diego, CA, USA). Statistical significance was evaluated by one-way Anova analysis, and $p<0.05$ was considered significant.

## Results

### ALA represses oxidative metabolism and glycolysis in ovarian cancer and TNBC cells

Since many tumors display the Warburg effect, many anti-cancer strategies aim to disrupt the aerobic glycolysis and promote oxidative phosphorylation. Heme has the potential to regulate glycolysis, TCA cycle, ETC, and OXPHOS. The first and rate-limiting step in heme synthesis is production of ALA [17]. Oral administration of the heme precursor ALA in mouse promotes aerobic energy metabolism in the liver [18]. In addition, ALA disrupts the Warburg effect and increases oxidative phosphorylation in human lung carcinoma cells [19].

To investigate whether stimulation of heme synthesis by ALA supplementation would affect the key metabolic pathways, we first determined heme levels in SKOV3 cells grown in the presence of ALA (Fig 1A). As expected, addition of the heme precursor ALA to cell culture medium elevated the cellular heme levels. Cell viability was not significantly affected by the used ALA doses; however, cell proliferation was significantly inhibited by ALA (Fig 1B). The increased heme levels were also reflected in dramatically upregulated heme oxygenase (HO-1) mRNA levels (Fig 1C). Transcription of catalase was also increased (Fig 1D), indicating that the increased cellular heme level in ALA-supplemented cells triggered increased synthesis of ROS, consistently with the role of free heme in ROS production [22]. ALA supplementation also reduced oxygen consumption rate (OCR) (Fig 1E) and extracellular acidification rate (ECAR) and lactate accumulation in the medium (Fig 1F). OCR reflects the activity of the ETC and OXPHOS. ECAR reflects production of lactic acid generated by aerobic glycolysis. ALA reduced basal respiration, as well as maximal respiration, and ATP production. ALA also reduced transcription of glycolytic genes glyceraldehyde-3-phosphate dehydrogenase (GAPDH) and hexokinase 2 (HK2) (Fig 1F). Despite of suppressing OCR, ALA supplementation elevated transcription of several ETC and OXPHOS genes (Fig 1G). In agreement with the role of Bach1 as a transcriptional activator of metastatic genes and a driver of cell migration [27–29], the expression of genes involved in metastasis and cell migration (MMP1, MMP9, MMP13, CXCR4, and vimentin), as well as cell migration, were reduced (Fig 1H and 1I).

Similarly to SKOV3 cells, ALA supplementation of TNBC cells MDA MB 231 also elevated heme levels (Fig 2A) without affecting cell viability. Cell proliferation was inhibited only at 10 mM concentration of ALA (Fig 2B). In comparison with SKOV3 cells, proliferation of MDA MB 231 cells was significantly less affected by ALA (compare Figs 1B and 2B). Elevated heme levels due to ALA supplementation significantly induced HO-1 transcription (Fig 2C) and slightly induced expression of catalase (Fig 2D). Again, similarly to SKOV3 cells, ALA supplementation of MDA MB 231 cells reduced OCR (Fig 2E) and ECAR and lactate accumulation in the medium (Fig 2F) and increased transcription of several ETC and OXPHOS genes (Fig

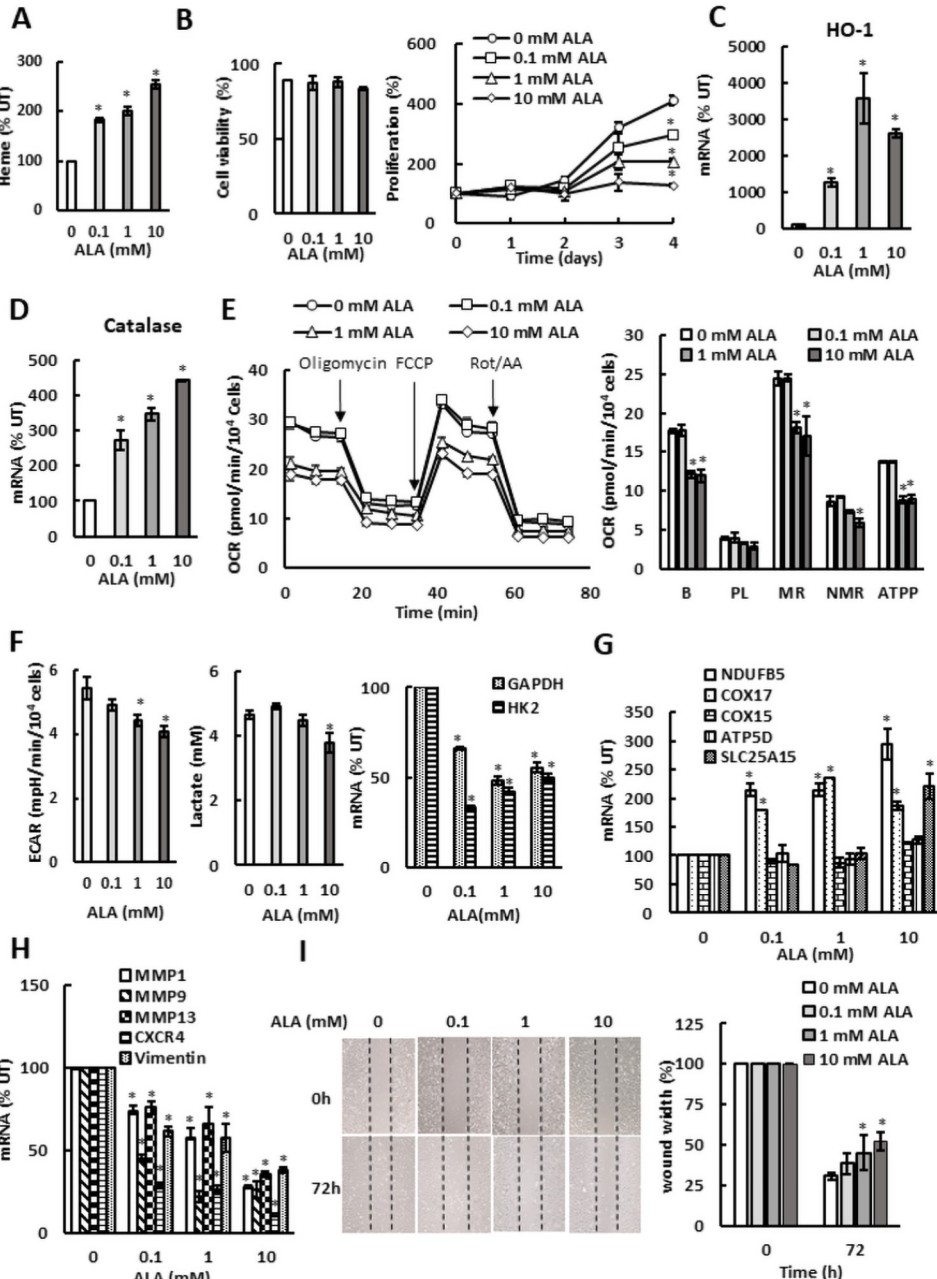

**Fig 1. ALA inhibits mitochondrial respiration and aerobic glycolysis in SKOV3 cells.** (A) Cellular heme levels in SKOV3 cells treated with ALA. Results are shown in comparison with untreated cells (%UT; 0 mM ALA). (B) Cell viability and proliferation. (C) HO-1 and (D) catalase mRNA analyzed by qRT-PCR in SKOV3 cells treated with ALA for 24 h. (E) Oxygen consumption rate (OCR) was measured in SKOV3 cells treated with ALA for 24 h. OCR was measured under basal conditions followed by the sequential addition of oligomycin (1.5 μM), FCCP (0.5 μM), and rotenone & antimycin A (Rot/AA; 0.5 μM) as indicated. Each data point represents an OCR measurement. Derived respiratory parameters are basal respiration (B), proton leak (PL), maximal respiration (MR), non-mitochondrial respiration (NMR), and ATP production (ATPP). (F) Extracellular acidification rate (ECAR), lactate accumulation in the medium, and GAPDH and HK2 mRNA levels in SKOV3 cells treated with ALA for 24 h. (I) Expression of metastatic genes in SKOV3 cells treated with ALA for 24 h. (G) Expression of ETC and OXPHOS genes measured by RT-PCR in SKOV3 cells treated with ALA for 24 h. (H) Expression of metastatic genes in SKOV3 cells treated with ALA for 24 h. (I) Migration of SKOV3 cells treated with ALA for 72 h analyzed by wound healing assay (n = 3, magnification: × 10). (A-I) The experiments were repeated three times, and the numerical results are shown as means ± S.D. Values that are statistically different ($p < 0.05$) from the control (untreated cells) are indicated by an asterisk.

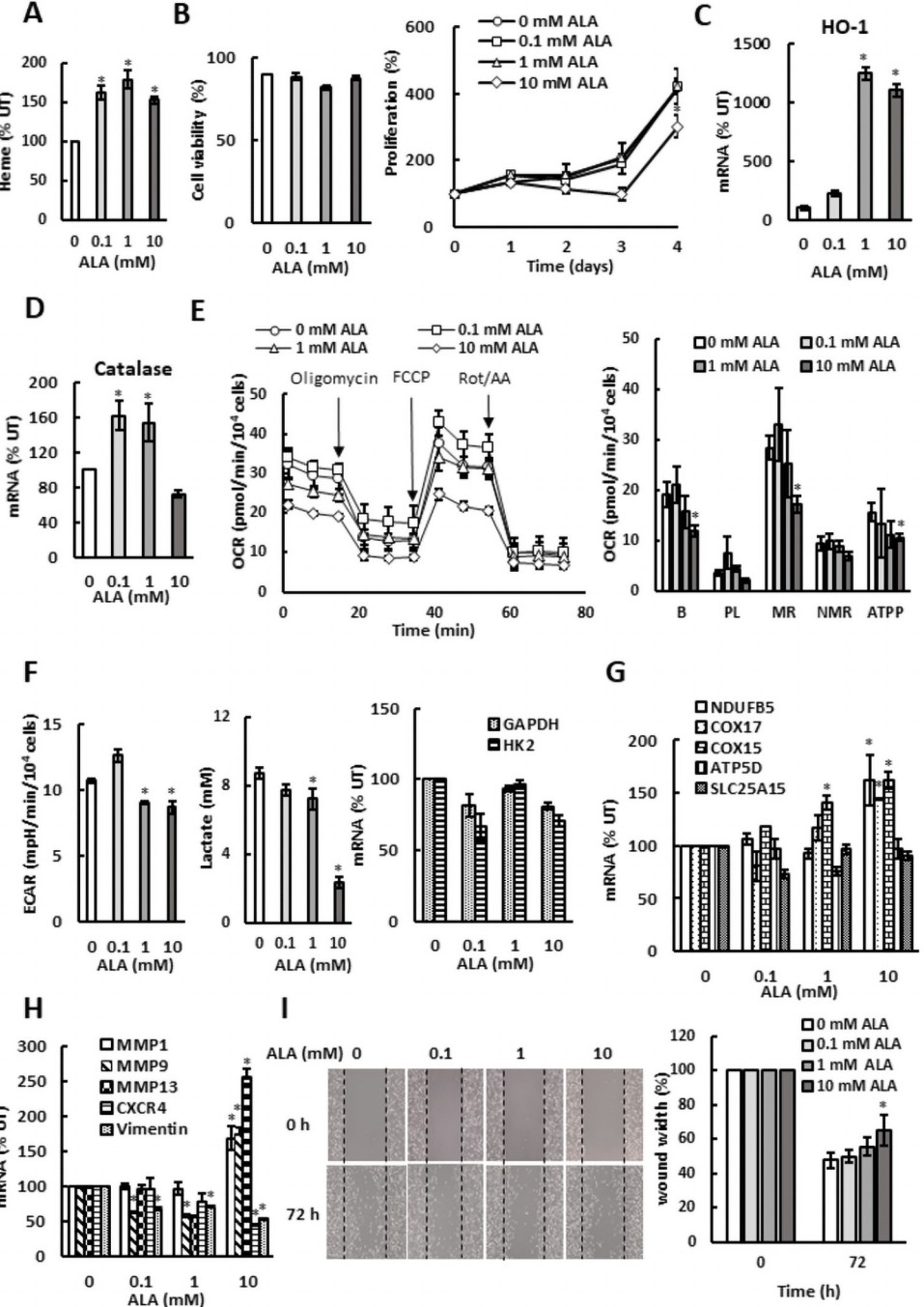

**Fig 2. ALA inhibits mitochondrial respiration and aerobic glycolysis in MDA MB 231 cells.** (A) Cellular heme levels and (B) cell viability and proliferation in MDA MB 231 cells treated with ALA. (C) HO-1 and (D) catalase mRNA analyzed by qRT-PCR in MDA MB 231 cells treated with ALA for 24 h. (E) OCR was measured in MDA MB 231 cells treated with ALA for 24 h. (F) ECAR, lactate accumulation in the medium, and GAPDH and HK2 mRNA levels in MDA MB 231 cells treated with ALA for 24 h. (G) Expression of ETC and OXPHOS genes in MDA MB 231 cells treated with ALA for 24 h. (H) Expression of metastatic genes in MDA MB 231 cells treated with ALA for 24 h. (I) Migration of MDA MB 231 cells treated with ALA for 72 h analyzed by wound healing assay (n = 3). (A-I) The experiments were repeated three times, and the numerical results are shown as means ± S.D. Values that are statistically different ($p < 0.05$) from the control (untreated cells) are indicated by an asterisk.

2G), while the mRNA levels of GADPH and HK2 were not significantly affected (Fig 2F). However, in comparison with SKOV3 cells, MDA MB 231 cells had more active glycolysis (compare Figs 1F and 2F). Despite elevating transcription of MMP1, MMP9, and MMP13 (Fig 2H), ALA supplementation slightly reduced cell migration (Fig 2I).

## ALA destabilizes Bach1 in SKOV3 and MDA MB 231 cells and activates AMPK in MDA MB 231 cells

Binding of heme to Bach1 inhibits its DNA binding activity and induces its nuclear export, resulting in polyubiquitination and degradation of Bach1 [30]. Consistently with the elevated heme levels, ALA supplementation reduced the Bach1 levels in both SKOV3 and MDA MB 231 cells (Fig 3A and 3B). Since supplementation with ALA reduced OCR and ECAR in both SKOV3 and MDA MB 231 cells, indicating downregulation of glycolysis and ETC/OXPHOS, two main energy-producing pathways, we determined the activation status of AMP-activated protein kinase (AMPK), an energy sensor and master regulator of metabolism. Activation of AMPK is associated with increased AMPK phosphorylation at Thr$^{172}$, a hallmark of AMPK activation by liver kinase B (LKB1) [36]. Another hallmark of AMPK activation is phosphorylation of acetyl-CoA carboxylase α (ACC) at Ser$^{79}$ [37]. Notwithstanding repression of both glycolysis and oxidative metabolism, supplementation with ALA did not result in increased phosphorylation of AMPK (pAMPK) or ACC (pACC) in SKOV3 cells (Fig 3C). However, ALA induced activation of AMPK in MDA MB 231 cells, as judged from increased phosphorylation of AMPK and ACC (Fig 3D). This is quite surprising, given that the proliferation of SKOV3 cells was significantly more affected by ALA than proliferation of MDA MB 231 cells.

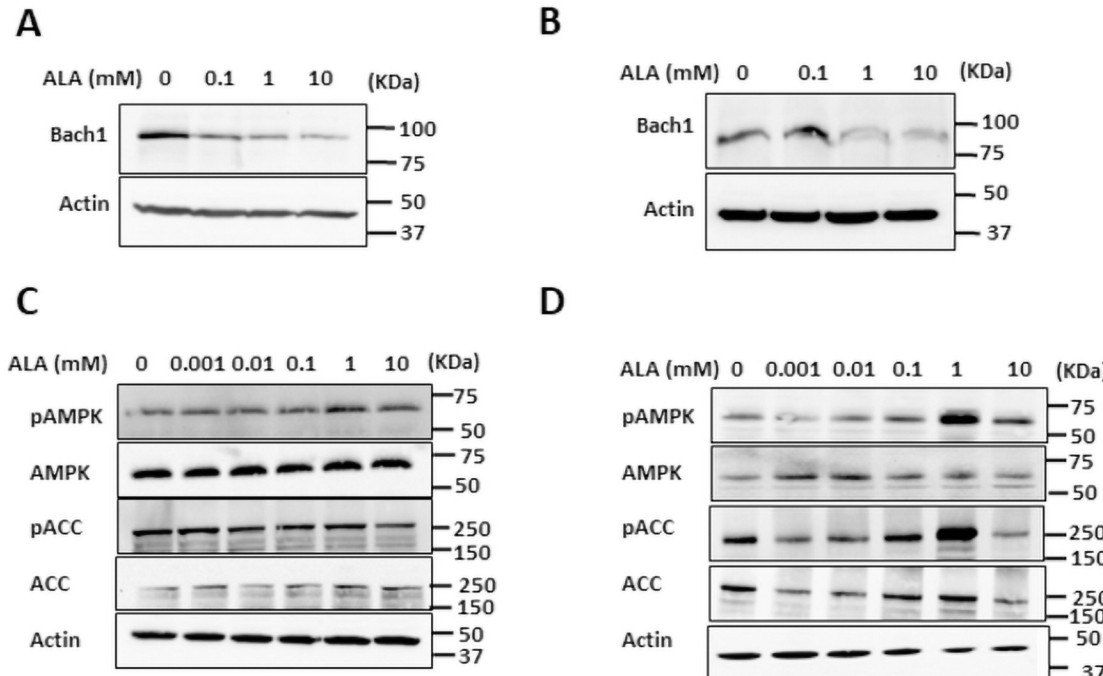

**Fig 3. ALA destabilizes Bach1 in SKOV3 and MDA MB 231 cells and activates AMPK in MDA MB 231 cells.** (A) SKOV3 and (B) MDA MB 231 cells were treated with 0, 0.1, 1, and 10 mM ALA for 24 h. Whole cell extracts were analyzed by Western blotting using antibodies against Bach1 and actin. (C) SKOV3 and (D) MDA MB 231 cells were treated with 0, 0.001, 0.01, 0.1, 1, and 10 mM ALA for 24 h and whole cell extracts were analyzed by Western blotting using antibodies against pAMPK, AMPK, pACC, ACC, and actin. The figures represent typical results from three independent experiments.

## AMPK represses oxidative metabolism and glycolysis in both ovarian cancer and TNBC cells

Despite repressing both glycolysis and oxidative metabolism, promoting heme synthesis by ALA supplementation did not activate AMPK in SKOV3 cells. However, treating SKOV3 cells with AMP homolog 5-amino-1-β-D-ribofuranosyl-1*H*-imidazole-4-carboxamide (AICAR) activated AMPK, as evidenced by increased levels of both pAMPK and pACC (Fig 4A). We conclude that AMPK in SKOV3 cells is competent for activation; however, SKOV3 cells likely have metabolic plasticity that allows them to adjust their metabolism to avoid the energy deprivation due to ALA treatment. As expected, treatment of SKOV3 cells with AICAR did not elevate cellular heme levels (Fig 4B). Interestingly, AICAR treatment paralleled the effects of ALA supplementation in both SKOV3 and MDA MB 231 cells and induced HO-1 and catalase transcription, albeit to a lesser degree than ALA supplementation (Fig 4C). AICAR treatment also reduced OCR (Fig 4D), ECAR and lactate accumulation in the medium (Fig 4E), as well as transcription of several metastatic genes (Fig 4F) and cell migration (Fig 4G). Similarly to SKOV3 cells, AICAR treatment of MDA MB 231 cells activated AMPK (Fig 5A) and did not elevate cellular heme levels (Fig 5B), but slightly induced HO-1 and catalase transcription (Fig 5C). AICAR treatment also reduced OCR (Fig 5D), ECAR and lactate accumulation in the medium (Fig 5E), as well as transcription of several metastatic genes and cell migration (Fig 5F and 5G). These results suggest that at least in MDA MB 231 cells, where AMPK is activated by ALA supplementation, AICAR treatment resulting in AMPK activation may contribute to the inhibitory effects of ALA on glycolysis, aerobic metabolism, and cell migration.

## Bach1 has a cell type-specific role in regulation of oxidative metabolism

Since increased levels of cellular heme by ALA supplementation reduced Bach1 protein levels, we wanted to determine whether Bach1 contributes to the regulation of glycolysis, aerobic metabolism, and cell migration. As shown in Fig 6A, small interfering RNA (siRNA)-mediated silencing of Bach1 suppressed the Bach1 mRNA and protein levels in SKOV3 cells by about 75%. Consistently with the role of Bach1 as a suppressor of HO-1, Bach1 silencing significantly elevated the HO-1 transcription (Fig 6B). Contrary to our expectation, however, Bach1 silencing reduced the OCR (Fig 6C), as well as mRNA levels of ETC genes NDUFB5 and COX17 (Fig 6D). In agreement with the role of Bach1 in transcription of metastatic genes, silencing of Bach1 reduced transcription of MMP9 and CXCR4, genes involved in metastasis and cell migration (Fig 6F). However, Bach1 silencing did not significantly affect migration of SKOV3 cells measured by the scratch assay (Fig 6G).

Bach1 silencing in TNBC cells MDA MB 231 significantly suppressed Bach1 protein and mRNA levels (Fig 7A) and induced expression of HO-1 (Fig 7B). Unlike in SKOV3 cells, Bach1 silencing in MDA MB 231 cells elevated OCR (Fig 7C). This is in agreement with the role of Bach1 in suppression of ETC genes in TNBC cells [26], but contrary to the role of Bach1 in regulation of oxidative metabolism in SKOV3 cells (Fig 6C). These findings suggest that the role of Bach1 in regulation of OCR is cell specific. Bach1 silencing in MDA MB 231 cells elevated mRNA level of only one of the several ETC genes examined (ATP5D; Fig 7D). Bach1 silencing in MDA MB 231cells also slightly elevated ECAR (Fig 7E). Contrary to the role of Bach1 as an activator of transcription of metastatic genes, silencing of Bach1 did not affect mRNA levels of MMP9, CXCR4, and vimentin, and elevated transcription of only MMP1 and MMP13 (Fig 7F).

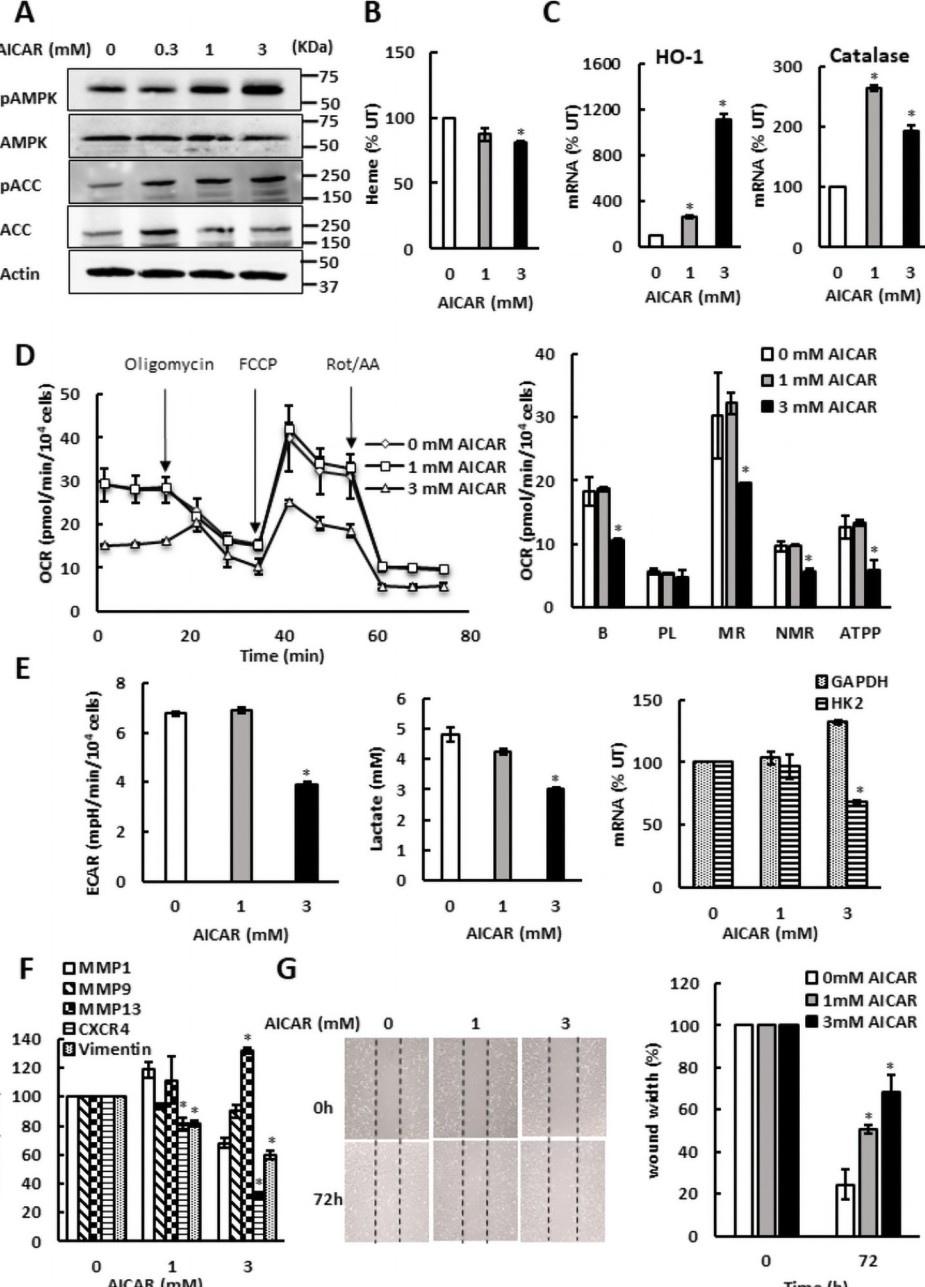

**Fig 4. AMPK inhibits mitochondrial respiration and aerobic glycolysis in SKOV3 cells.** (A) SKOV3 cells were treated with 0, 0.3, 1, and 3 mM AICAR for 24 h and whole cell extracts were analyzed by Western blotting using antibodies against pAMPK, AMPK, pACC, ACC, and actin. The figure represents typical results from three independent experiments. (B) Cellular heme levels in SKOV3 cells treated with AICAR. (C) HO-1 and catalase mRNAs were analyzed by qRT-PCR in SKOV3 cells treated with AICAR for 24 h. (D) OCR was measured in SKOV3 cells treated with AICAR for 24 h. (E) ECAR, lactate accumulation in the medium, and GAPDH and HK2 mRNA levels in SKOV3 cells treated with AICAR for 24 h. (F) Expression of metastatic genes in SKOV3 cells treated with AICAR for 24 h. (G) Migration of SKOV3 cells treated with AICAR for 72 h analyzed by wound healing assay (n = 3). (A-G) The experiments were repeated three times, and the numerical results are shown as means ± S.D. Values that are statistically different ($p < 0.05$) from the control (untreated cells) are indicated by an asterisk.

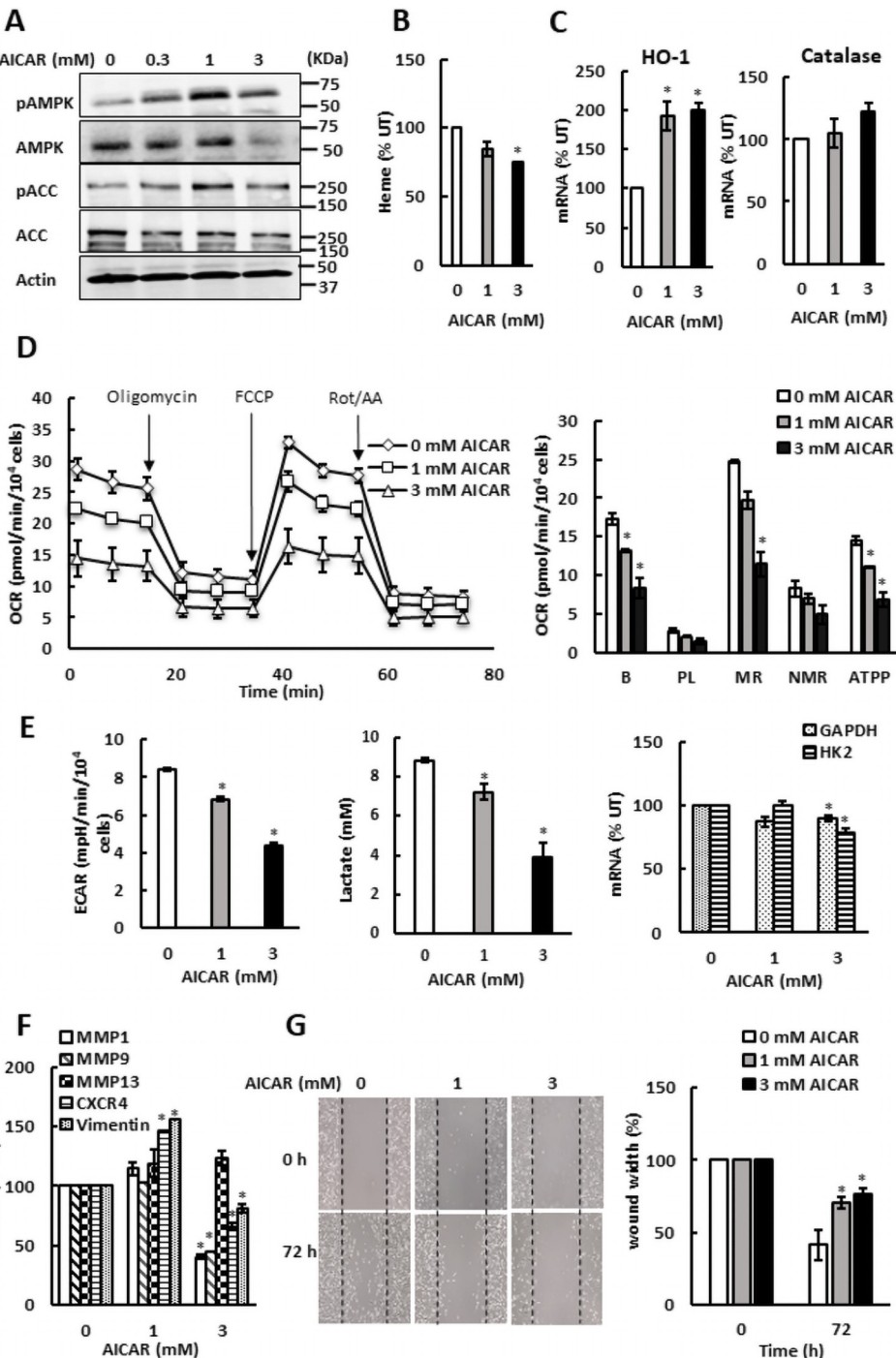

**Fig 5. AMPK inhibits mitochondrial respiration and aerobic glycolysis in MDA MB 231 cells.** (A) MDA MB 231 cells were treated with 0, 0.3, 1, and 3 mM AICAR for 24 h and whole cell extracts were analyzed by Western blotting using antibodies against pAMPK, AMPK, pACC, ACC, and actin. The figure represents typical results from three independent experiments. (B) Cellular heme levels in MDA MB 231 cells treated with AICAR for 24 h. (C) HO-1 and catalase mRNAs were analyzed by qRT-PCR in MDA MB 231 cells treated with AICAR for 24 h. (D) OCR was measured in MDA MB 231 cells treated with AICAR for 24 h. The experiment was performed as described in Fig 1E. (E) ECAR, lactate accumulation in the medium, and GAPDH and HK2 mRNA levels in MDA MB 231 cells treated with AICAR for 24 h. (F) Expression of metastatic genes in MDA MB 231 cells treated with AICAR for 24 h. (G) Migration of MDA MB 231 cells treated with AICAR for 72 h analyzed by wound healing assay (n = 3). (A-G) The experiments were repeated three times, and the numerical results are shown as means ± S.D. Values that are statistically different ($p < 0.05$) from the control (untreated cells) are indicated by an asterisk.

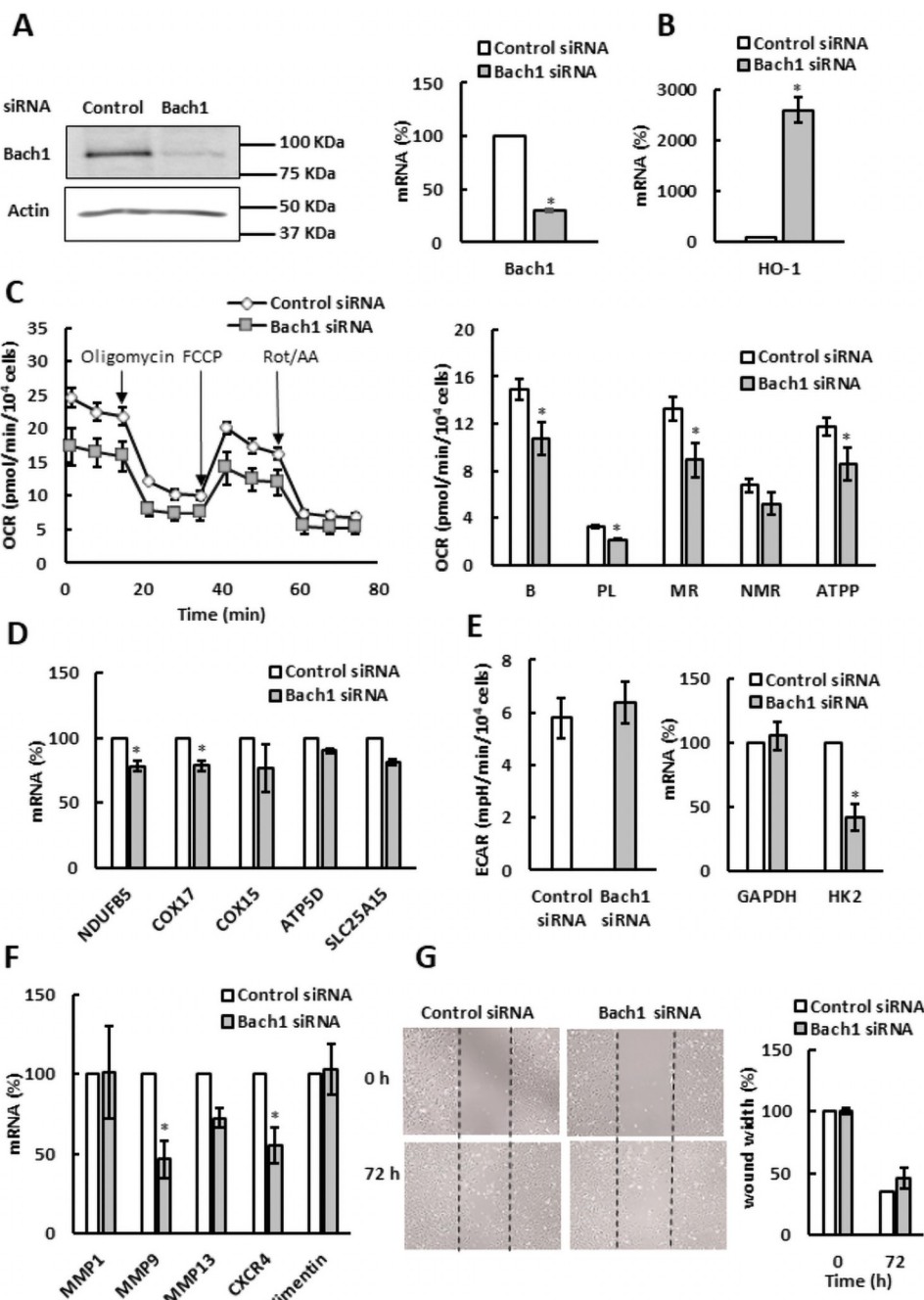

**Fig 6. Bach1 regulates mitochondrial respiration in SKOV3 cells.** SKOV3 cells were transfected with Bach1 or control siRNA and analyzed by (A) western blotting using Bach1 and actin antibodies and qRT-PCR. (B) HO-1 mRNA levels, (C) OCR, (D) expression of ETC and OXPHOS genes, (E) ECAR and GAPDH and HK2 mRNA levels, (F) expression of metastatic genes, and (G) migration of SKOV3 cells analyzed by wound healing assay (n = 3). (A-G) The experiments were repeated three times, and the numerical results are shown as means ± S.D. Values that are statistically different ($p < 0.05$) from the control (SKOV3 cells transfected with control siRNA) are indicated by an asterisk.

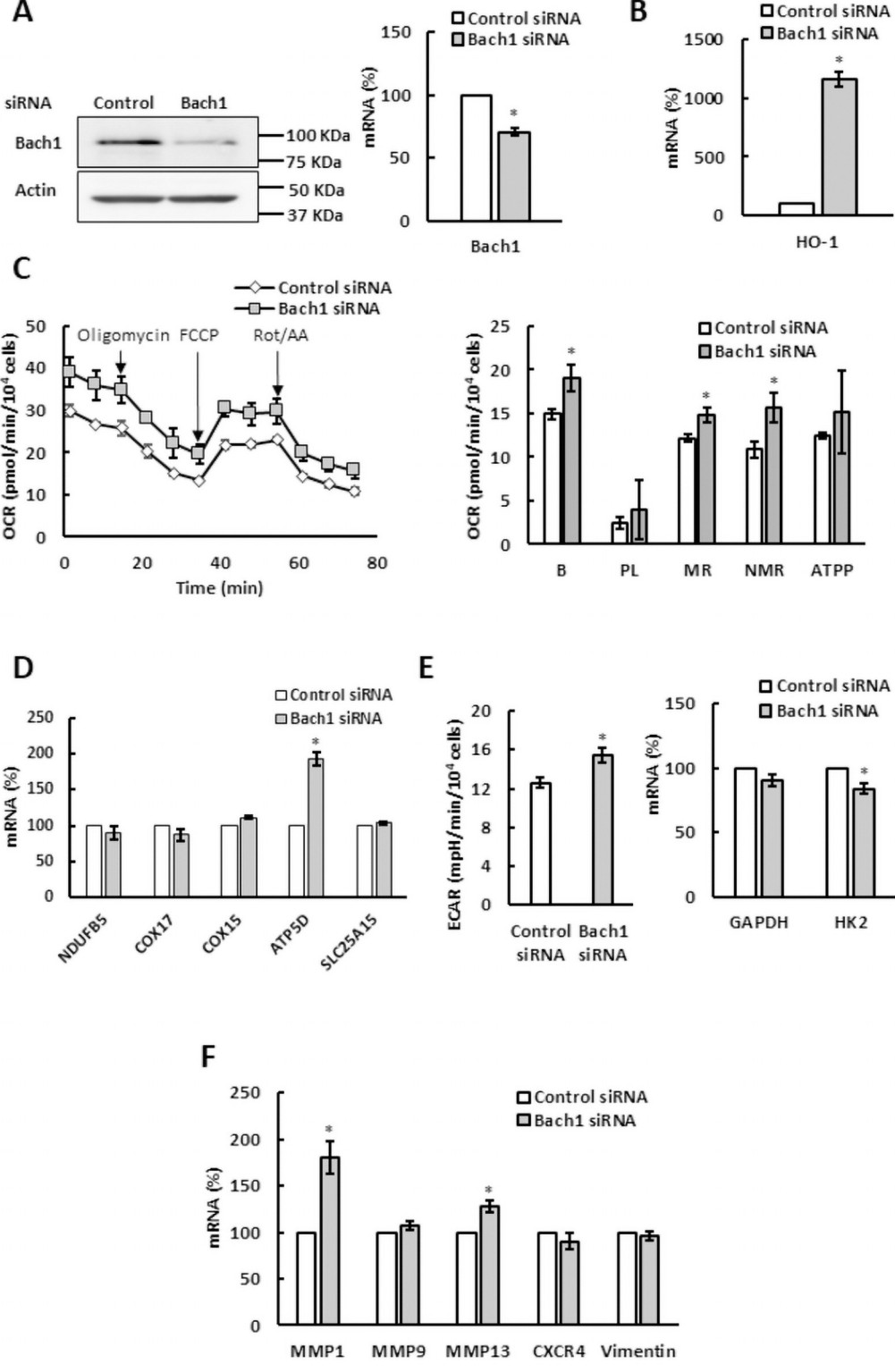

**Fig 7. Bach1 regulates mitochondrial respiration in MDA MB 231 cells.** MDA MB 231 cells were transfected with Bach1 or control siRNA and analyzed by (A) western blotting using Bach1 and actin antibodies and qRT-PCR. (B) HO-1 mRNA levels, (C) OCR, (D) expression of ETC and OXPHOS genes, (E) ECAR and GAPDH and HK2 mRNA levels, and (F) expression of metastatic genes. (A-F) The experiments were repeated three times, and the numerical results are shown as means ± S.D. Values that are statistically different ($p < 0.05$) from the control (MDA MB 231 cells transfected with control siRNA) are indicated by an asterisk.

## Cells with different activities of aerobic metabolism display unique patterns of ALA sensitivity

In comparison with SKOV3 cells, MDA MB 231 cells displayed slightly higher activity of aerobic metabolism (compare Figs 1E and 2E) and their proliferation was significantly less affected by ALA treatment (compare Figs 1B and 2B). To evaluate whether the activity of oxidative metabolism correlates with ALA sensitivity, we compared the OCR and proliferation in several cell lines with different activities of oxidative metabolism (Fig 8). Breast cancer MCF7 cells and lung cancer H1299 cells displayed very high OCR and their proliferation was not significantly affected by 0.1 mM and 1 mM ALA concentrations. However, their proliferation was completely inhibited by 10 mM ALA. Despite having lower OCR than MCF7 and H1299 cells, proliferation of prostate cancer PC3 cells at different ALA concentrations displayed similar pattern and was only slightly affected at 0.1 mM and 1 mM ALA; however, at 10 mM, the proliferation of PC3 cells was almost completely inhibited (Fig 8). In contrast to MCF7 and H1299 cells, MDA MB 468 cells displayed lower OCR and their proliferation was affected by all ALA concentrations. Remarkably, MDA MB 468 cells were able to proliferate even in the presence of 10 mM ALA and the OCR values and the proliferation pattern of MDA MB 468 cells resembles that of MDA MB 231 cells. The ovarian cancer OVCAR3 cells behaved similarly as SKOV3 cells (compare Figs 1B and 1E with 8) and displayed the lowest OCR. Similarly to MDA MB 468 cells, their proliferation was affected by all ALA concentrations and OVCAR3 cells were able to proliferate even in the presence of 10 mM ALA.

## Discussion

The key finding of this study is that promoting heme synthesis through ALA supplementation in both ovarian cancer and TNBC cells leads to downregulation of glycolysis, mitochondrial respiration, as well as cell proliferation. In comparison with MDA MB 231 cells, SKOV3 cells displayed lower activities of both glycolytic and aerobic metabolism (compare Figs 1E and 1G with 2E and 2G) and their proliferation was significantly more suppressed by ALA treatment, especially at low ALA concentrations (compare Figs 1B and 2B). The underlying mechanisms involve destabilization of Bach1, AMPK activation, and inducing anti-oxidant response. This result suggests that cells that rely more heavily on aerobic metabolism for energy production are less sensitive to ALA. To test this notion, we evaluated response to ALA in several cell lines with different activities of oxidative metabolism (Fig 8). The general conclusion of these studies is that proliferation of cells with a high activity of oxidative metabolism, such as H1299 and MCF7, is not affected by low concentrations of ALA (0.1 and 1 mM), but significantly inhibited at high ALA concentration (10 mM). In contrast, proliferation of cells with a lower activity of oxidative metabolism, such as MDA MB 468 and OVCAR3, is affected even by low (0.1 and 1 mM) ALA concentrations. The inverse correlation between the OCR and ALA sensitivity is particularly interesting for cells with the same origin, as shown for MCF7, MDA MB 231, and MDA MB 468 (Figs 2 and 8). Moreover, SKOV3 and OVCAR3 cells have lower OCR and are more sensitive to ALA than MDA MB 231, and MDA MB 468 cells (Figs 1, 2 and 8).

Heme serves as a prosthetic group in proteins that sense, transport, or use oxygen, such as hemoglobin, myoglobin, cytochrome complexes, catalases, and cyclooxygenases. Heme is also required for ATP production by the mitochondrial ETC and OXPHOS and controls activities of a number of proteins involved in regulation of metabolism and transcription [17]. One of heme's cellular targets is Bach1. Bach1 is a transcriptional regulator that controls expression of several groups of genes important for metabolism and metastasis of cancer cells. In lung cancer, administration of antioxidants N-acetylcysteine or vitamin E reduces level of free heme, which stabilizes Bach1. Bach1, in turn, activates HK2 and GAPDH and stimulates glycolysis-

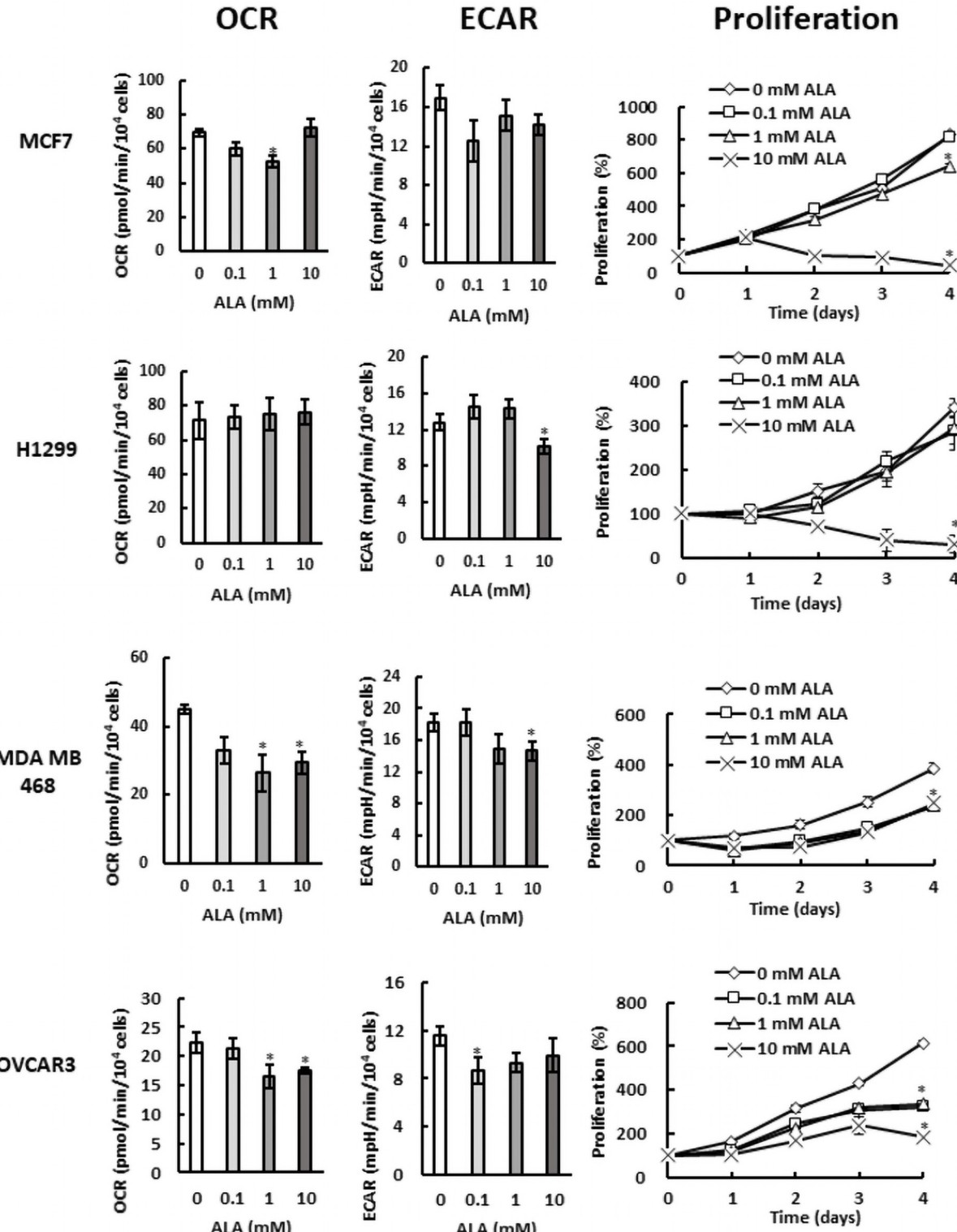

**Fig 8. Cells with different activities of aerobic metabolism display unique patterns of ALA sensitivity.** OCR, ECAR, and proliferation in breast cancer MCF7 cells, lung cancer H1299 cells, TNBC cells MDA MB 468, and ovarian cancer OVCAR3 cells.

dependent metastasis [24]. About 30% of non-small-cell lung cancers display increased expression of antioxidant genes due to acquiring mutations that stabilize Nrf2, a transcription factor that regulates expression of antioxidant genes. The elevated activity of Nrf2 leads to increased expression of HO-1 and degradation of heme, resulting in stabilization of Bach1 and increased transcription of pro-metastatic genes [25]. The level of Bach1 is also elevated in triple-negative breast cancer (TNBC), leading to lower activity of TCA cycle and decreased transcription of ETC genes. Suppressing or destabilizing Bach1 renders TNBC cells sensitive to ETC inhibitors, such as metformin [26]. These reports highlight that Bach1 targeting is a possible strategy for cancer treatment.

Activating heme synthesis by exogenous administration of ALA provides an effective way to regulate Bach1 stability (Fig 3A and 3B). In SKOV3 cells, ALA supplementation paralleled the effects of Bach1 silencing (Fig 6) and reduced ETC and OXPHOS (Fig 1). These results indicate that Bach1 is a positive regulator of ETC and OXPHOS in SKOV3 cells and likely mediates the effect of increased level of heme on aerobic metabolism. In MDA MB 231 cells, ALA supplementation reduced ETC, OXPHOS, and glycolysis (Fig 2). However, Bach1 silencing had the opposite effect and activated aerobic metabolism (Fig 7). We conclude that the effect of elevated level of heme on ETC, OXPHOS, and glycolysis in MDA MB 231 cells (Fig 2) is at least partly mediated by the activation of AMPK elicited by ALA supplementation (Figs 3D and 5). The opposite effects of Bach1 silencing on aerobic metabolism in SKOV3 and MDA MB 231 cells (Figs 6 and 7) suggest that the role of Bach1 in regulation of oxidative metabolism depends on the context and is rather cell-type specific. To determine to what extent the regulation of aerobic metabolism by ALA supplementation is dependent or independent of Bach1will require *in vitro and in vivo* studies utilizing Bach1 mutants that are not regulated by heme [26, 38].

As indicated by the increased expression of anti-ROS gene catalase in both SKOV3 and MDA MB 231 cells (Figs 1 and 2), elevated heme levels due to ALA supplementation lead to increased production of ROS [22, 39] and may contribute to the suppression of oxidative metabolism in both cell lines (Figs 1 and 2). ALA treatment of human hepatocarcinoma cells induced ROS production and inhibited mitochondrial respiration [23]. Modulation of heme synthesis by exogenous administration of ALA is used in photodynamic therapy (PDT) of cancer. ALA induces elevated levels of protoporphyrin IX (PPIX), the ultimate precursor in the heme biosynthetic pathway [40], primarily in cancer cells. PPIX is an efficient photosensitizer and activation of PPIX by light generates ROS in cancer cells, leading to apoptotic cell death [41–43]. The role of oxidative stress and ROS in tumorigenesis is controversial and many results support the role of ROS in promoting or suppressing tumorigenesis [44–46]. Thus, while our results show that targeting Bach1 by exogenous administration of ALA is quite effective in reducing Bach1 levels, the additional effects of ALA likely involve elevated ROS production. Promoting heme synthesis by ALA supplementation may thus represent a promising new anti-cancer strategy, particularly in cancers that are sensitive to altered redox signaling, or in combination with strategies that target the antioxidant systems or metabolic weaknesses of cancer cells.

## Supporting information

**S1 File.**
(PDF)

## Author Contributions

**Conceptualization:** Pritpal Kaur, Yan Zhu, Ivana Vancurova, Ales Vancura.

**Data curation:** Pritpal Kaur, Ales Vancura.

**Formal analysis:** Pritpal Kaur, Ales Vancura.

**Funding acquisition:** Ales Vancura.

**Investigation:** Pritpal Kaur, Shreya Nagar, Madhura Bhagwat, Mohammad Uddin, Ales Vancura.

**Methodology:** Pritpal Kaur, Yan Zhu, Ivana Vancurova, Ales Vancura.

**Project administration:** Ivana Vancurova, Ales Vancura.

**Resources:** Yan Zhu, Ivana Vancurova, Ales Vancura.

**Supervision:** Ivana Vancurova, Ales Vancura.

**Validation:** Pritpal Kaur, Ales Vancura.

**Visualization:** Pritpal Kaur.

**Writing – original draft:** Pritpal Kaur, Ales Vancura.

**Writing – review & editing:** Ivana Vancurova, Ales Vancura.

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
