## [Decision Letter · Decision Letter 0]

10 Mar 2021

PONE-D-20-40407

Activated heme synthesis regulates glycolysis and oxidative metabolism in breast and ovarian cancer cells

PLOS ONE

Dear Dr. Vancura,

Thank you for submitting your manuscript to PLOS ONE. After careful consideration, we feel that it has merit but does not fully meet PLOS ONE’s publication criteria as it currently stands. Therefore, we invite you to submit a revised version of the manuscript that addresses the points raised during the review process.

I believe the findings are very interesting and would certainly be strengthened by testing in additional TNBC cell lines. 

We look forward to receiving your revised manuscript.

Kind regards,

Pankaj K Singh, Ph.D.

Academic Editor

PLOS ONE

Reviewers' comments:

Reviewer's Responses to Questions

**Comments to the Author**

1. Is the manuscript technically sound, and do the data support the conclusions?

Reviewer #1: Yes

Reviewer #2: Yes

2. Has the statistical analysis been performed appropriately and rigorously? 

Reviewer #1: Yes

Reviewer #2: Yes

3. Have the authors made all data underlying the findings in their manuscript fully available?

Reviewer #1: Yes

Reviewer #2: Yes

4. Is the manuscript presented in an intelligible fashion and written in standard English?

Reviewer #1: Yes

Reviewer #2: Yes

5. Review Comments to the Author

Reviewer #1: This is an interesting paper that investigates the role of Heme levels and cell metabolism and migration. Key roles for the transcription factor BACH1 and AMPK. The experiments are convincing and the conclusions are supported by the presented experiments. The material and methods are adequately described. The analysis is limited to just two cell lines, MDA-MB-231 and SKOV3 cells which limits impact of these studies and the metabolic analysis is limited to measurements of ECAR and OCR. In addition, the response of the two cell lines to ALA is not the same, leaving one to wonder about the universality of the author’s principal findings.

Based on the experiments where AICAR is used to activate AMPK, the authors state several times in the results section that AMPK has certain activities in their different assays. It would be more correct to state that AICAR treatment has a particular activity and corelate that activity with AMPK activation state. In essence, the experiments don’t adequately test AMPK necessity.

Reviewer #2: This article demonstrates that enhancing heme synthesis through exogenous supplementation of heme precursor 5-aminolevuliic acid can suppresses oxygen consumption as well as glycolysis and reduces proliferation of ovarian cancer and TNBC cells. Such findings imply a possibility for 5-aminolevuliic acid to treat cancer. The study seems to be an interesting. However, the idea is not so novel as the similar studies with other cancer cell lines have been described. Some justification on the originality of the work is necessary. Specific critiques are provided below:

1. The authors performed extracellular acidification rate (ECAR) approach only to test glycolytic rate. This is limited in that ECAR is highly related to pH value of cell medium and CO2 production. The standard measurements of glycolytic metabolism are the use of 3H labeled glucose and lactate concentration in culture medium of cells treated with or without drugs.

2. In all figures, mRNA levels of HO-1 and Catalase have been tested. However, as important enzymes in the catabolism of heme, the protein levels of these enzymes should be examined as well.

3. Justification of literature citations regarding the dosages of ALA are missing.

4. In figure 1F and 2F, could you explain why you choose these targets (NDUFB5, cox17, cox15, ATP5D and SLC25A15) as ETC and OXPHOS genes, instead of coxⅣ, coxⅤ or any other standard genes?

5. In figure 2I, the area of wound width is not in appropriate position.

6. The data indicated that Bach1 was reduced by ALA supplementation, and silencing Bach1 could reduce ECT and OXPHOS, which is similar to ALA supplementation treatment. However, whether Bach1 is involved in heme for reducing ETC, OXPHOS, and glycolysis, was not discussed. Does any approach (such as chemicals) block the degradation of Bach1 in ALA supplemented cells? And after that, further ETC, OXPHOS, and glycolysis analysis should be performed.

7. The lack of vivo data to validate the results in culture is another limitation of the research, which should be discussed.

6. PLOS authors have the option to publish the peer review history of their article (what does this mean?). If published, this will include your full peer review and any attached files.

Reviewer #1: No

Reviewer #2: **Yes: **Wei Wang

---

## [Author Response · Author response to Decision Letter 0]

4 Aug 2021

Response to Reviewers

Academic Editor:

1. “I believe the findings are very interesting and would certainly be strengthened by testing in additional TNBC cell lines.”

Response: We agree; we have included additional four cell lines, including another TNBC cell line (MDA MB 468) and another breast cancer cell line (MCF7) (Fig. 8 of the revised manuscript).

Reviewer 1:

1. “This is an interesting paper that investigates the role of Heme levels and cell metabolism and migration. Key roles for the transcription factor BACH1 and AMPK. The experiments are convincing and the conclusions are supported by the presented experiments. The material and methods are adequately described. The analysis is limited to just two cell lines, MDA‐MB‐231 and SKOV3 cells which limits impact of these studies and the metabolic analysis is limited to measurements of ECAR and OCR. In addition, the response of the two cell lines to ALA is not the same, leaving one to wonder about the universality of the author’s principal findings.”

Response: We have included additional four cell lines with different activities of oxidative metabolism (Fig. 8 of the revised manuscript). The general conclusion of these studies is that proliferation of cells with a high activity of oxidative metabolism, such as H1299 and MCF7, is not affected by low concentrations of ALA (0.1 and 1 mM). In contrast, proliferation of cells with a lower activity of oxidative metabolism, such as SKOV3 and OVCAR3, is affected even by low (0.1 and 1 mM) ALA concentrations. The inverse correlation between the OCR and ALA sensitivity is particularly interesting for cells with the same origin, as shown for MCF7, MDA MB 231, and MDA MB 468 (Fig. 2 and 8 of the revised manuscript).

2. “Based on the experiments where AICAR is used to activate AMPK, the authors state several times in the results section that AMPK has certain activities in their different assays. It would be more correct to state that AICAR treatment has a particular activity and corelate that activity with AMPK activation state. In essence, the experiments don’t adequately test AMPK necessity.”

Response: We agree; we have corrected the Results section “AMPK represses oxidative metabolism and glycolysis in both ovarian cancer and TNBC cells” (p. 12 of the revised manuscript).

Reviewer 2:

“This article demonstrates that enhancing heme synthesis through exogenous supplementation of heme precursor 5‐aminolevuliic acid can suppresses oxygen consumption as well as glycolysis and reduces proliferation of ovarian cancer and TNBC cells. Such findings imply a possibility for 5‐aminolevuliic acid to treat cancer. The study seems to be an interesting. However, the idea is not so novel as the similar studies with other cancer cell lines have been described. Some justification on the originality of the work is necessary.”

Response: We agree; we have modified the last paragraph of the Introduction to better explain the originality of the work. (“In this study, we explored the role of cellular heme level in regulation of Warburg effect and expression of pro-metastatic genes in ovarian cancer and TNBC cells. Additional motivation for this study was to determine whether supplementation with heme precursor 5-aminolevulinic acid (ALA) downregulates Bach1. We show that ALA supplementation leads to downregulation of glycolysis, mitochondrial respiration, as well as cell migration. The underlying mechanisms differ in ovarian and TNBC cells, but likely involve destabilization of Bach1, activation of AMP-activated protein kinase (AMPK), and induction of anti-oxidant response.”)

1. “The authors performed extracellular acidification rate (ECAR) approach only to test glycolytic rate. This is limited in that ECAR is highly related to pH value of cell medium and CO2 production. The standard measurements of glycolytic metabolism are the use of 3H labeled glucose and lactate concentration in culture medium of cells treated with or without drugs.”

Response: We agree; we have determined the lactate concentration in the culture medium. The new results are included in Fig. 1, 2, 4, and 5.

2. “In all figures, mRNA levels of HO‐1 and Catalase have been tested. However, as important enzymes in the catabolism of heme, the protein levels of these enzymes should be examined as well.”

Response: Since heme is known to regulate HO-1 and catalase at the transcriptional level, we believe that measuring mRNA levels of HO-1 and catalase is the most direct and informative way to test the effect of ALA.

4. “In figure 1F and 2F, could you explain why you choose these targets (NDUFB5, cox17, cox15, ATP5D and SLC25A15) as ETC and OXPHOS genes, instead of coxⅣ, coxⅤ or any other standard genes?”

Response: These ETC and OXPHOS genes (NDUFB5, cox17, cox15, ATP5D and SLC25A15) encode components of electron transport chain and mitochondrial ATP synthase. Cox15, ATP5D and SLC25A15 were identified as targets of Bach1 (Lee et al., Effective breast cancer combination therapy targeting Bach1 and mitochondrial metabolism. Nature 568, 254-258, 2019). Since one of the possible mechanisms of ALA supplementation and heme levels is mediated by Bach1, we believe that selecting these genes is appropriate. NDUFB5 and cox17 were identified as targets of LKB1-AMPK-PGC1α axis and were designed on the basis of the corresponding paper (Li et al., CAB39L elicited an anti-Warburg effect via a LKB1-AMPK-PGC1α axis to inhibit gastric tumorigenesis. Oncogene 37:6383–6398, 2018).

5. “In figure 2I, the area of wound width is not in appropriate position.”

Response: We agree and we corrected Fig. 2I.

6. “The data indicated that Bach1 was reduced by ALA supplementation, and silencing Bach1 could reduce ECT and OXPHOS, which is similar to ALA supplementation treatment. However, whether Bach1 is involved in heme for reducing ETC, OXPHOS, and glycolysis, was not discussed. Does any approach (such as chemicals) block the degradation of Bach1 in ALA supplemented cells? And after that, further ETC, OXPHOS, and glycolysis analysis should be performed.”

Response: The only known approach to stabilize Bach1 is inhibition of the proteasome by MG132 (Zenke-Kawasaki et al., Heme induces ubiquitination and degradation of the transcription factor Bach1. Mol. Cell. Biol. 27, 6962-6971, 2007) or other proteasome inhibitors, such as Bortezomib. However, inhibition of proteasome (in ALA treated or control cells) would produce a pleiotropic effect that would be difficult to interpret.

7. “The lack of vivo data to validate the results in culture is another limitation of the research, which should be discussed.”

Response: We modified the Discussion section of the manuscript to indicate the need for in vivo studies of ALA supplementation.

I would like to thank the Reviewers for the useful critiques and suggestions.

Ales Vancura

---

## [Decision Letter · Decision Letter 1]

10 Nov 2021

Activated heme synthesis regulates glycolysis and oxidative metabolism in breast and ovarian cancer cells

PONE-D-20-40407R1

Dear Dr. Vancura,

We’re pleased to inform you that your manuscript has been judged scientifically suitable for publication and will be formally accepted for publication once it meets all outstanding technical requirements.

Kind regards,

Pankaj K Singh, Ph.D.

Academic Editor

PLOS ONE

Additional Editor Comments (optional):

Reviewers' comments:

Reviewer's Responses to Questions

**Comments to the Author**

1. If the authors have adequately addressed your comments raised in a previous round of review and you feel that this manuscript is now acceptable for publication, you may indicate that here to bypass the “Comments to the Author” section, enter your conflict of interest statement in the “Confidential to Editor” section, and submit your "Accept" recommendation.

Reviewer #1: All comments have been addressed

2. Is the manuscript technically sound, and do the data support the conclusions?

Reviewer #1: Yes

3. Has the statistical analysis been performed appropriately and rigorously? 

Reviewer #1: Yes

4. Have the authors made all data underlying the findings in their manuscript fully available?

Reviewer #1: Yes

5. Is the manuscript presented in an intelligible fashion and written in standard English?

Reviewer #1: Yes

6. Review Comments to the Author

Reviewer #1: Thank you for addressing my concerns. No further comments. In my opinion this manuscript makes an interesting contribution to the literature

7. PLOS authors have the option to publish the peer review history of their article (what does this mean?). If published, this will include your full peer review and any attached files.

Reviewer #1: **Yes: **Don Ayer

---

## [Editor Report · Acceptance letter]

12 Nov 2021

PONE-D-20-40407R1 

Activated heme synthesis regulates glycolysis and oxidative metabolism in breast and ovarian cancer cells 

Dear Dr. Vancura:

I'm pleased to inform you that your manuscript has been deemed suitable for publication in PLOS ONE. Congratulations! Your manuscript is now with our production department. 

Kind regards, 

on behalf of

Dr. Pankaj K Singh 

Academic Editor

PLOS ONE